# TimewarpVAE: Simultaneous Time-Warping and Representation Learning of Trajectories

## Abstract

Human demonstrations of trajectories are an important source of training data for many machine learning problems. However, the difficulty of collecting human demonstration data for complex tasks makes learning efficient representations of those trajectories challenging. For many problems, such as for handwriting or for quasistatic dexterous manipulation, the exact timings of the trajectories should be factored from their spatial path characteristics. In this work, we propose TimewarpVAE, a fully differentiable manifold-learning algorithm that incorporates Dynamic Time Warping (DTW) to simultaneously learn both timing variations and latent factors of spatial variation. We show how the TimewarpVAE algorithm learns appropriate time alignments and meaningful representations of spatial variations in small handwriting and fork manipulation datasets. Our results have lower spatial reconstruction test error than baseline approaches and the learned low-dimensional representations can be used to efficiently generate semantically meaningful novel trajectories.

## 1 Introduction

Continuous trajectories are inherently infinite-dimensional objects that can vary in complex ways in both time and space. However, in many practical situations, they contain intrinsic sources of variability that can be well-approximated by projection onto a low-dimensional manifold. For instance, when a human demonstrates trajectories for a robot, it is useful for the robot to learn to model the most expressive latent factors controlling the spatial paths of the demonstration trajectories. For certain types of demonstrations, such as in gesture control or quasistatic manipulation, it is highly advantageous to explicitly separate the exact timing of the trajectory from the spatial latent factors.

As an illustrative example, consider trying to average two samples from a handwriting dataset generated by humans drawing the letter "A" in the air (Chen et al., 2012). If we scale two trajectories linearly in time so that both their timestamps go from 0 to 1, and then average the two trajectories at each timestep, the resulting average does not maintain the style of the "A"s. This is because the average is taken between parts of the two trajectories that do not naturally correspond to each other. An example of this averaging, with lines showing examples of points that are averaged, is shown in Fig. 1a. A common approach like Dynamic Time Warping (DTW) (Sakoe & Chiba, 1978) can lead to unintuitive results. When averaging these same two trajectories. DTW only takes in information about these two trajectories, and does not use contextual information about other examples of the letter "A" to better understand how to align the timings of these trajectories[1] In Fig. 1b we use the `dtw` package (Giorgino, 2009) to align the trajectories before averaging them at corresponding timesteps. We see that the resulting trajectory is spatially close to the input trajectories, but it again does not maintain the style of the "A"s.

Our TimewarpVAE takes the time alignment benefits of DTW and uses them in a manifold learning algorithm to align the timings of similar trajectories. Our results are shown in Fig. 1d. Our approach is similar to the Rate Invariant Autoencoder of Koneripalli et al. (2020), which also learns a latent space that separates timing and spatial factors of variation, with the spatial interpolations shown in Fig. 1c. Our TimewarpVAE approach has two main contributions which make it better suited for generating robot trajectories compared to a Rate Invariant Autoencoder. These contributions

---

[1] We use the terms "time warping," "time alignment," and "registration" interchangeably.

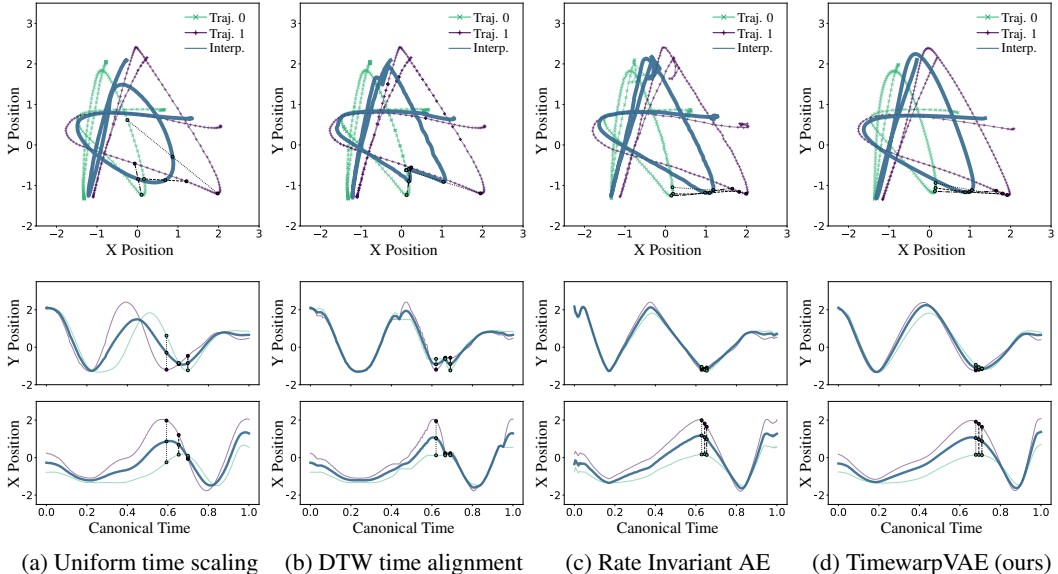

Figure 1: Interpolations in latent space between canonical trajectories, using various models. For Rate Invariant Autoencoder and TimewarpVAE, we use a sixteen dimensional spatial latent space and the interpolation is constructed by decoding the average of the spatial latent embeddings. The resulting average trajectory is plotted alongside the reconstructions of the original two trajectories. The Rate Invariant Autoencoder can learn to ignore parts of the canonical trajectory during training, leading to the jittering seen at the beginning and end of the canonical trajectory.

include the parameterization of the generated trajectory as an arbitrarily complicated neural network function of time, rather than basing the trajectory calculation on piecewise linear functions with a pre-specified number of knots. The second is that our approach includes a regularization term so that the model is properly penalized for extreme time warps. Combined, these additions prevent the model from learning to ignore poorly generated sections of its canonical trajectories, which is a problem seen at the beginning and end of the trajectories in Fig. 1c. Additionally, we use a VAE architecture instead of a simple autoencoder.

## 2 RELATED WORK

Learning a latent space of trajectories that combines together the timing and spatial parameters into a single latent space appears in (Coll-Vinent & Vondrick, 2022), (Chen et al., 2021), and (Lu et al., 2019) among others. TimewarpVAE, like the Rate Invariant Autoencoder of Koneripalli et al. (2020), instead separates the timing and spatial latent variables, giving a more efficient spatial model. Our work improves on the Rate Invariant Autoencoder in two important ways. The first is that our work is not constrained to learning a piecewise linear trajectory. and the second is that we include a regularization term to penalize the timewarper from trying to ignore parts of the template trajectory. Comparing our results to the Rate Invariant Autoencoder shows that this timing regularization is important to combat a degeneracy in the choice of timing of the canonical trajectory. We explain this degeneracy in Section 4.5 and explain how it also applies to the work of Shapira Weber et al. (2019) in Appendix A.3.

Functional Data Analysis (Wang et al., 2016) involves the study of how to interpret time-series data, often requiring the registration of data with different timings. The idea of warping the timings of paths appears in, for example, Dynamic Time Warping (Sakoe & Chiba, 1978) and the calculation of the Fréchet distance (Fréchet, 1906). Our work is most closely related to Continuous Dynamic Time Warping (Kruskal & Liberman, 1983), which we refine for the manifold-learning setting. The registration and averaging of trajectories is performed by Petitjean et al. (2011), Schultz & Jain (2018), and Williams et al. (2020). Rather than just learn an average trajectory, we model the full manifold of trajectories. Time-warping is used by Chang et al. (2021) to learn "discriminative

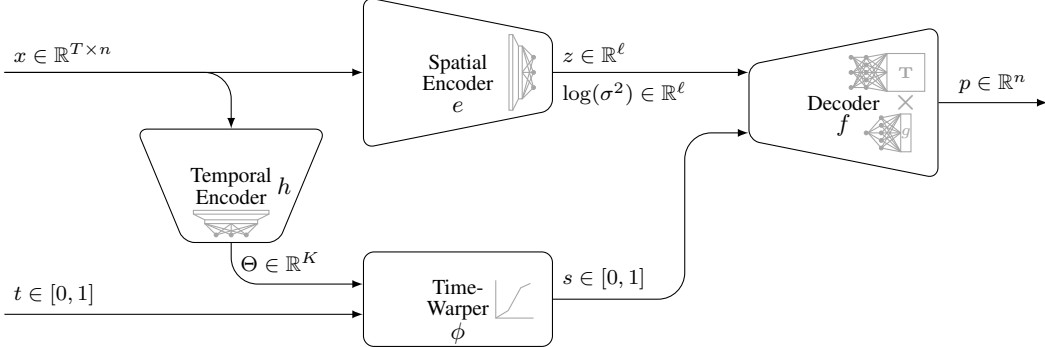

Figure 2: The architecture for TimewarpVAE. TimewarpVAE takes in a full trajectory $z$ and a timestamp $t$, and reconstructs the position $p$ of the trajectory at that timestamp. TimewarpVAE separately encodes the timing of the trajectory into $\Theta$ and encodes the spatial information into a latent distribution parameterized by $z$ and $\log(\sigma^2)$.

prototypes" of trajectories, but not a manifold representation of the trajectories. A linear model to give a learned representation of registered trajectories is generated in Kneip & Ramsay (2008), and our work can be considered an expansion of that work, now using manifold learning to allow for nonlinear spatial variations of registered trajectories.

Time-warping has previously been combined with manifold learning to generate representations of individual frames of a trajectory. For example, Zhou & la Torre (2012), Trigeorgis et al. (2018), and Cho et al. (2023) align trajectories and learn representations of individual frames contained in the trajectories. Connectionist Temporal Classification (CTC) can also be viewed as an algorithm for learning a labelling of frames of a trajectory while ignoring timing variations (Graves et al., 2006). Instead, our approach focuses on learning a latent vector representation that captures information about the entire trajectory.

There are many different ways that trajectory data can be parameterized when presented to the learning algorithm, for example, the trajectory could be parameterized as a Dynamic Movement Primitive (DMP) (Ijspeert et al., 2013) before learning a (linear) model of DMP parameters, as is done in Matsubara et al. (2011). You can also learn a DMP over a learned representation of states (Chen et al., 2016). Modeling trajectories using DMPs can be useful; for example, you can vary the timing during execution to allow a robot to "catch up" and correct for execution errors (Schaal et al., 2007). However, that work does not model timing variations during training. TimewarpVAE accounts for timing variations during training, enabling its latent variable to concentrate its modeling capacity on spatial variations of trajectories.

## 3 APPROACH

A standard approach to learning a manifold for trajectories (see, for example the method proposed by Chen & Müller (2012)) is to map each trajectory to a learned representation that includes information about timing and spatial variations. This type of representation learning can be performed by a beta-Variational Auto-Encoder (beta-VAE) (Higgins et al., 2017). We provide a brief introduction to beta-VAE for comparison with our TimewarpVAE.

### 3.1 BETA-VAE

A beta-VAE encodes each datapoint $x$ into a learned probability distribution $q(z|x)$ in the latent space and decodes points in the latent space into probability distributions $p(x|z)$ in the data space. A standard formulation of beta-VAE is to parameterize the encoder distributions as axis-aligned Gaussians $q(z|x) = \mathcal{N}(e(x), \sigma^2(x))$. Thus, the encoder returns the expected latent value $z = e(x) \in \mathbb{R}^\ell$ for a trajectory, along with the log of the (diagonal) covariance of the encoder noise distribution given by $\log(\sigma^2) \in \mathbb{R}^\ell$. For continuous data, we constrain the decoder distribution to have spherical covariance with diagonal elements all equal to some constant (not-learned) value $\sigma_R^2$.

Thus, the decoder $f$ only needs to return the expeced decoded trajectory $\tilde{x}$, given a latent value. The beta-VAE architecture is shown in Fig. 7. The training objective is described in Appendix A.1.

## 3.2 TimewarpVAE

TimewarpVAE is based on beta-VAE, with the goal of concentrating modeling capacity on spatial variations. In order to separate the spatial and temporal variations in trajectories, TimewarpVAE contains two additional modules, not present in beta-VAE: a temporal encoder and a time-warper. The decoder now takes in information from both the spatial encoder and the time-warper. Fig. 2 shows the architecture of TimewarpVAE. It takes in two inputs, the training trajectory $x$, and a desired reconstruction time $t$. Like in beta-VAE, the spatial encoder maps the trajectory $x$ to its latent value distribution parameterized by $z$ and $\log(\sigma^2)$. The temporal encoder computes time-warping parameters $\Theta$, and the time-warper (defined by the parameters $\Theta$) now acts on $t$ to warp it to a "canonical time" $s$. These modules are explained in Sections 4.1 and 4.2. The decoder takes the canonical time $s$ and the spatial latent vector and returns the position of the canonical trajectory for that latent vector at the canonical time $s$. All these should be trained so that the decoded position is a good reconstruction of the position of trajectory $x$ at timestep $t$, while at the same time minimizing the total information that we allow the autoencoder to store about the trajectory.

Specifically, the minimization objective for TimewarpVAE, denoted $\mathcal{L}$, is the sum of the reconstruction cost $\mathcal{L}_R$, beta-VAE's KL divergence loss $\mathcal{L}_{\text{KL}}$, and a new time-warping regularization $\mathcal{L}_\phi$, which we explain further in Section 4.5: $\mathcal{L} = \mathcal{L}_R + \mathcal{L}_{\text{KL}} + \mathcal{L}_\phi$, where

$$\mathcal{L}_R = \frac{1}{\sigma_R^2} \mathbb{E}_{x_i, t, \epsilon} \left[ \left\| x_i(t) - f\left( \sum_{j=1}^{k} h(x_i)_j \psi_j(t),\, e(x_i) + \epsilon \right) \right\|^2 \right] \tag{1}$$

$$\mathcal{L}_{\text{KL}} = \beta \mathbb{E}_{x_i} \left[ \frac{\|e(x_i)\|^2 + \sum_d (\sigma_d^2(x_i) + \log(\sigma_d^2(x_i)))}{2} \right] \tag{2}$$

$$\mathcal{L}_\phi = \lambda \mathbb{E}_{x_i} \left[ \frac{1}{K} \sum_{j=1}^{K} (h(x_i)_j - 1) \log(h(x_i)_j) \right] \tag{3}$$

For clarity, we again use the subscript $x_i$, to emphasize that these losses are computed as empirical expectations over each of the training trajectories $x_i$. $e$, $\sigma^2$ (and its elements $\sigma_d^2$), $\epsilon$, and $\sigma_R^2$ are all defined in the same way as for beta-VAE in Eq. 6. $f$ is the decoder, now taking in a canonical timestamp as well as the latent value. $h$ is the temporal encoder, so that $h(x_i)_j$ is the $j$th output neuron (out of $K$ total) of the temporal encoder applied to the $i$th trajectory. The $\psi_j$ are the time-warping basis functions, explained in Sec. 4.1 and defined in Equation 4. $\beta$ is a regularization hyperparameter for beta-VAEs. $\lambda$ is a regularization hyperparameter for our time-warping functions. Since $\lambda$ penalizes the timewarping in the model, a large $\lambda \to \infty$ would drive time-warping to be the identity function, and $\lambda = 0$ could allow the model to learn severe time warps. We explain the neural network implementation of all of these functions in the next section.

The benefits of this algorithm are as follows: it learns a low-dimensional representation of spatial variations in trajectories; it can be implemented as a neural-network and trained using backpropagation; and it can accommodate non-linear timing differences in the training trajectories. Additionally, you can generate new trajectories using a latent value $z$ and canonical timestamps $s$ ranging from 0 to 1 without using the time-warper or the temporal encoder, which we do in our empirical evaluations, calling these generated trajectories "canonical" or "template" trajectories. These trajectories outperform baseline qualitatively in Fig. 1d and quantitatively in our results section.

## 4 Neural Network Formulation

In this section, we explain how to write the spatial encoding function, the temporal encoding function, the time-warping function, and the decoding function as differentiable neural networks. The time-warping function is a differentiable neural network with no learnable parameters, since the time-warping is entirely defined by the input parameters $\Theta$. The other three modules have learnable parameters which we learn through backpropagation.

### 4.1 NEURAL NETWORK ARCHITECTURE FOR THE TIME-WARPER

The time-warper takes in a training timestamp $t$ for a particular trajectory and maps it monotonically to a canonical timestamp $s$. We formulate $\phi$ as a piecewise linear function of $t$ with equally spaced knots and $K$ linear segments. We label the slopes of those segments $\Theta_j$ for $1 \leq j \leq K$. Different choices of vector $\Theta \in \mathbb{R}^K$ give different time-warping functions. In order for $\Theta$ to yield a valid time-warping function mapping $[0, 1]$ to $[0, 1]$, the $\Theta_j$ should be positive and average to 1. These $\Theta$ values are generated by the temporal encoding function discussed in the next section. Given some vector $\Theta$, corresponding to the slope of each segment, the time-warper $\phi$ is given by $\phi(t) = \sum_{j=1}^{K} \Theta_j \psi_j(t)$ where the $\psi_j$ are defined by the following and do not need to be learned:

$$\psi_j(t) = \min \left\{ \max \left\{ t - (j-1)/K, 0 \right\}, 1/K \right\} \tag{4}$$

A visualization of these basis functions $\psi_j$ is presented in Fig. 8 in Appendix A.4. We use the specific parameterization of $\Theta_j$ described in the next section to ensure that our time-warping function is a bijection from $[0, 1]$ to $[0, 1]$.

### 4.2 NEURAL NETWORK ARCHITECTURE FOR THE TEMPORAL ENCODER

We use a neural network $h : \mathbb{R}^{n \times T} \to \mathbb{R}^K$ to compute a different vector $\Theta$ for each training trajectory $x$. To ensure the elements of $\Theta$ are positive and average to 1, we take the softmax of the last layer of the temporal encoder and scale the result by $K$. This transformation sends the values $\theta$ to $\Theta_j = \text{Softmax}(\theta)_j K$ for $j$ from 1 to $K$, ensuring that the average of the output neurons $\Theta_j$ is 1 as desired. By contrast, Koneripalli et al. (2020) square the last layer and then normalize.

### 4.3 NEURAL NETWORK ARCHITECTURE FOR THE SPATIAL ENCODER

Given a trajectory $x$ evenly sampled at $T$ different timesteps $t_j$ between 0 and 1, we write the $T \times n$ matrix of these evenly sampled positions $x(t_j)$ as $x \in \mathbb{R}^{n \times T}$. In the neural network architecture used in our experiments, one-dimensional convolutions are applied over the time dimension, treating the $n$ spatial dimensions as input channels. This is followed by different fully connected layers for $e$ and for $\log(\sigma)$. However, any neural network architecture, such as a Transformer (Vaswani et al., 2017) or Recurrent Neural Network (Hochreiter & Schmidhuber, 1997), could be used in the spatial encoder module of a TimewarpVAE.

### 4.4 NEURAL NETWORK ARCHITECTURE OF THE DECODER

Any architecture that takes in a time $s$ and a latent vector $z$ and returns a position $p$ could be used for the decoder $f$. We choose to use a modular decomposition of $f$, with $f(s, z)$ decomposed as the product of a matrix and a vector: $f(s, z) = \mathbf{T}(z)g(s)$. In this formulation, the matrix $\mathbf{T}(z)$ is a function of the latent vector $z$ and the vector $g(s)$ is a function of the (retimed) timestep $s$. If each point in the training trajectory has dimension $n$, and if we choose a hyperparameter $m$ for our architecture, the matrix $\mathbf{T}(z)$ will have shape $n \times m$, and the vector $g(s)$ will be of length $m$. We compute the $nm$ elements of $\mathbf{T}(z)$ as the (reshaped) output of a sequence of fully-connected layers taking the vector $z$ as an input. The $m$ elements of $g(s)$ are computed as the output of a sequence of fully-connected layers taking the scalar $s$ as an input. Because we know that the scalar $s$ will lie in the range $[0, 1]$, we customize the initialization of the weights in the first hidden layer of $g(s)$. Details are provided in Appendix A.10. We choose this architecture for the benefit of the "NoNonlinearity" ablation experiment in Section 5.5. With this architecture, if we ablate by removing the hidden layers in $\mathbf{T}(z)$, making it a linear function of $z$, then entire decoder becomes linear with respect to $z$, so our generated position $p$ will be a linear combination of possible positions at timestep $s$.

### 4.5 REGULARIZATION OF TIME-WARPING FUNCTION

The choice of timing for the canonical trajectories adds a degeneracy to the solution.[2] Without this regularization, it is possible for other methods, like Rate Invariant Autoencoders, to warp so severely

---

[2]This is similar to the degeneracy noted in Appendix A.2.1 for Continuous Dynamic Time Warping (Kruskal & Liberman, 1983), and, as noted in Appendix A.3, was not properly analyzed in (Shapira Weber et al., 2019).

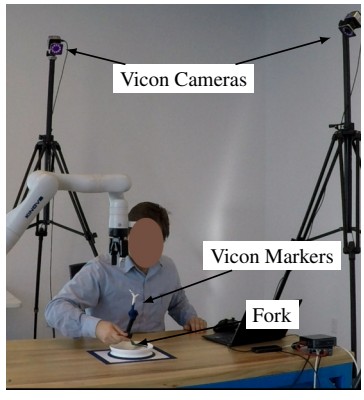
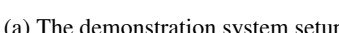

(a) The demonstration system setup

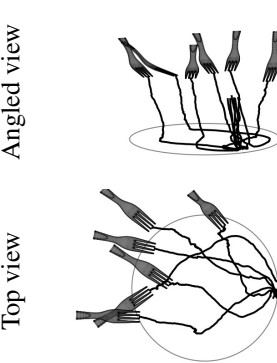

(b) Six example demonstrations[3]

Figure 3: We collect trajectory recordings of the position and orientation of a fork while it is used to pick a small piece of yarn off a plate with steep sides. Example trajectories are shown from two angles by showing the initial orientation of the fork and the position of the tip of the fork over time.

that they can learn to ignore parts of the canonical trajectory. This is a problem if we are generating robot motions based on the canonical trajectory, since it can lead to jittering motions, as seen at the beginning and end of the learned trajectory in Fig. 1c.

We propose a regularization penalty on the time-warper $\phi$ to choose among the degenerate solutions. We penalize $\int_0^1 (\phi'(t) - 1) \log(\phi'(t)) \, dt$. That regularization contains the function $g(x) = (x - 1)\log(x)$ applied to the slope $\phi'(t)$, integrated over each point $t$. That function $g(x)$ is concave-up for $x > 0$ and takes on a minimum at $x = 1$, thus encouraging the slope of $\phi$ to be near 1. This function has the nice symmetric property that regularizing $\phi(t)$ using this function gives the exact same regularization as regularizing $\phi^{-1}(s)$. This symmetry is proven in Appendix A.8. For each trajectory $x_i$ our time-warper $\phi$ is stepwise linear with $K$ equally-sized segments with slopes $\Theta_1 = h(x_i)_1, \ldots, \Theta_k = h(x_i)_K$. Thus, the regularization integral for the time-warper associated with $x_i$ is

$$\mathcal{L}_\phi(x_i) = \frac{1}{K} \sum_{j=1}^{K} (h(x_i)_j - 1) \log(h(x_i)_j) \tag{5}$$

## 5 EXPERIMENTS

We perform experiments on two datasets, one containing handwriting gestures made in the air while holding a Wii remote (Chen et al., 2012), and one which we collect ourselves of motions to pick yarn up off of a plate with a fork. We use the same model architecture for both experiments, with hyperparameters given in Appendix A.13. Additionally, during training, we perform data augmentation by randomly perturbing the timesteps used when sampling the trajectory $x$, using linear interpolation to compute sampled positions. Our specific data-augmentation implementation is described in Appendix A.11. This data augmentation decreases training performance, but greatly improves test performance, as shown in the ablation studies in Section 5.5.

### 5.1 FORK TRAJECTORY DATASET

We record 345 fork trajectories using a Vicon tracking system shown in Fig. 3a. Reflective markers are rigidly attached to a plastic fork, and the trajectory of the fork is recorded using Vicon cameras. A six-centimeter length of yarn is placed on the plate in an arbitrary location and orientation. It is then picked up in a right-handed motion by scraping it to the left and using the side of the plate

---

[3]Fork meshes in 3D trajectory plots were downloaded and modified from https://www.turbosquid.com/3d-models/metal-fork-3d-model/362158 and are used under the TurboSquid 3D Model License

Table 1: Performance Results for 3-dimensional models of fork trajectories. Our TimewarpVAE significantly outperforms beta-VAE and the ablation of TimewarpVAE without the time-warper.

| Architecture | Beta | Rate | Training Aligned RMSE ($\pm 3\sigma$) | Test Aligned RMSE ($\pm 3\sigma$) |
|---|---|---|---|---|
| TimewarpVAE | 0.01 | 3.716 | $0.187 \pm 0.003$ | $\mathbf{0.233 \pm 0.003}$ |
| | 0.1 | 3.227 | $\mathbf{0.185 \pm 0.007}$ | $0.234 \pm 0.008$ |
| RateInvariantAE | 0.01 | 4.095 | $0.260 \pm 0.130$ | $0.316 \pm 0.188$ |
| | 0.1 | 3.280 | $0.285 \pm 0.154$ | $0.325 \pm 0.132$ |
| beta-VAE | 0.01 | 4.759 | $0.291 \pm 0.005$ | $0.343 \pm 0.016$ |
| | 0.1 | 3.670 | $0.293 \pm 0.007$ | $0.342 \pm 0.011$ |
| NoTimewarp | 0.01 | 3.924 | $0.264 \pm 0.007$ | $0.360 \pm 0.017$ |
| | 0.1 | 3.508 | $0.265 \pm 0.006$ | $0.354 \pm 0.014$ |

as a static tool to push it onto the fork. Demonstrations were intentionally collected with three different timings, where in some trajectories the approach to the plate was intentionally much faster than the retreat from the plate, in some trajectories the approach was intentionally much slower than the retreat from the plate, and in the remaining demonstrations the approach and retreat are approximately the same speed. The dataset was split into 240 training trajectories and 105 test trajectories. Examples of six recorded trajectories, along with visualizations of the starting pose of the fork for those trajectories are presented in Fig. 3b. Trajectories are truncated to start when the fork tip passes within 10cm of the table and to end again when the fork passes above 10cm. All trajectories were subsampled to 200 equally-spaced time points, using linear interpolation as needed. We express the data as the $x, y, z$ position of the tip of the fork and the $rw, rx, ry, rz$ quaternion orientation of the fork, giving a dataset of dimension $n = 7$ at each datapoint. We preprocess the data to choose the sign of the quaternion representations to all near each other in $\mathbb{R}^4$. We mean-center the data by subtracting the average $x, y, z, rw, rx, ry, rz$ training values, and we divide the $x, y, z$ values by a normalizing factor so that their combined variance $\mathbb{E}[x^2 + y^2 + z^2]$ is 3 on the training set. We multiply the $rw, rx, ry, rz$ values by a scaling factor of $0.08m$, (chosen as a length scale associated with the size of the fork), before dividing by the same normalizing factor, to bring all the dimensions into the same range. We make this dataset publicly available in our supplemental materials.

## 5.2 HANDWRITING GESTURES DATASET

For the handwriting experiment, we use the air-handwriting dataset that was made by Chen et al. (2012). We project the drawn letters onto xy coordinates, which makes the data easy to visualize. We take a training set of 125 random examples of the letter "A" drawn from the air-handwriting dataset, and we take the remaining 125 random examples of the letter "A" as our test set. All trajectories were subsampled to 200 equally-spaced time points, using linear interpolation as needed. We also mean-center so that the average position over the whole training dataset is the origin, and scale $x$ and $y$ together by the same constant so that their combined variance $\mathbb{E}[x^2 + y^2]$ is 2 on the training set. Example training trajectories are shown in Fig. 9.

## 5.3 MODEL PERFORMANCE MEASURES

To classify performance, we inspect three important measures: the training reconstruction error, the test reconstruction error, and the model rate. Since we are interested in the ability of our model to measure spatial variation of trajectories, we compute reconstruction errors by first performing symmetric DTW to align the reconstructed trajectory to the original trajectory. We then compute the Euclidean mean squared error between each point in the original trajectory and every point it is paired with. After that, we calculate the average of all those errors over all the timesteps in the original trajectory before taking the square root to compute our aligned root mean squared error (Aligned RMSE). In the framework of Rate-Distortion theory, these errors are distortion terms. The model rate is a measure of the information bottleneck imposed by the model, and is given by the KL divergence term in the beta-VAE loss function (Alemi et al., 2017). It's important to check the rate of the model since arbitrarily complex models can get perfect reconstruction error if they have a large enough rate (Alemi et al., 2018). However, among trained models with similar rates, it is fair to say that the model with lower distortion is a better model. We report the model rate in bits.

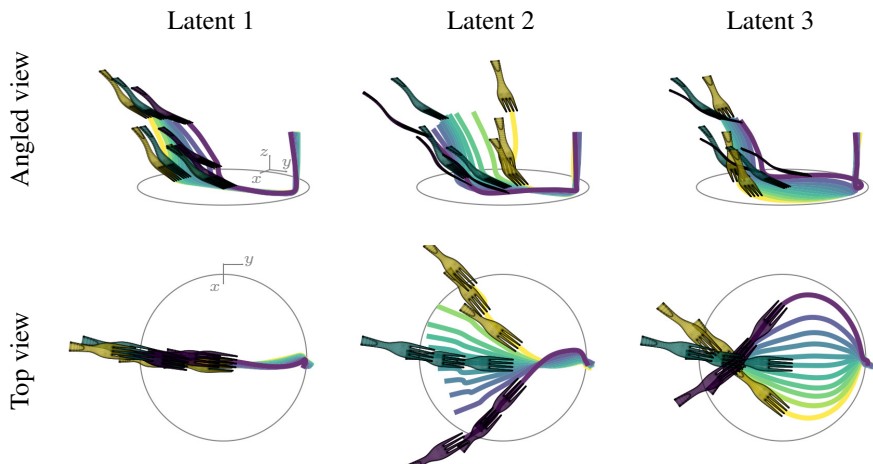

Figure 4: The paths of the fork tip are plotted over time for trajectories generated by TimewarpVAE for different latent vectors. The orientation of the fork is shown at three different timesteps in a color matching the color of its associated path. Each latent dimension has an interpretable effect. The first latent dimension determines the fork's initial $y$ position, the second determines the fork's initial $x$ position, and the third determines how the fork curves during trajectory execution.

## 5.4 FORK MODEL RESULTS

We train models with a latent dimension of three on the fork dataset. We compare TimewarpVAE to beta-VAE and to an ablation of TimewarpVAE called "NoTimewarp" which sets our time-warping module to the identity function. The latent sweep of a TimewarpVAE trained with $\beta = 1$ is shown in Fig. 4. Performance measures are shown in Table 1, where we compute and summarize five trials for each model type, for various hyperparameters $\beta$. TimewarpVAE outperforms the baseline methods and learns an interpretable latent space.

## 5.5 AIR HANDWRITING RESULTS

We compare TimewarpVAE to beta-VAE and a VAE version of Rate Invariant Autoencoders, trained on the same training and test splits. We train all models with a batch size of 64 for 20,000 epochs, using the Adam optimizer and a learning rate 0.001. We train each model five times each for different choices of beta, and we plot the mean along with one standard error above and below the mean. To give context to these results, we also show results for Parametric Dynamic Movement Primitives (Matsubara et al., 2011), and PCA results, which do not have associated rate values. TimewarpVAE significantly outperforms Parametric DMPs, PCA, and beta-VAE, with the greatest differences for smaller latent dimensions. TimewarpVAE outperforms Rate Invariant Autoencoders for low dimensions. For higher dimensions, Rate Invariant Autoencoders perform slightly better in training error and comparably in test error, which may be due to differences in model architectures.

We run ablation studies to understand the importance of our architecture choices. The first is to remove the data augmentation that adds additional trajectories with perturbed timings. We call this ablation "NoAugment." The second is to remove the hidden layers in the neural network $\mathbf{T}(z)$. Removing the hidden layers makes $\mathbf{T}(z)$ a linear function, meaning that the function $f$ can only learn trajectories that are a linear function of the latent variable $z$ (but $f$ is still nonlinear in the time argument $s$). We call this ablation "NoNonlinearity". The third is to remove the time-warper and replace it with the identity function. We call this ablation "NoTimewarp." Results for these ablations are plotted in Fig. 6. NoAugment confirms the importance of data augmentation for good generalization. NoAugment is able to get a slightly better fit to the training data than TimewarpVAE, but performs poorly on test data. NoNonlinearity has comparable performance on the test data to TimewarpVAE, showing that the strict regularization imposed by its linearity condition is good for

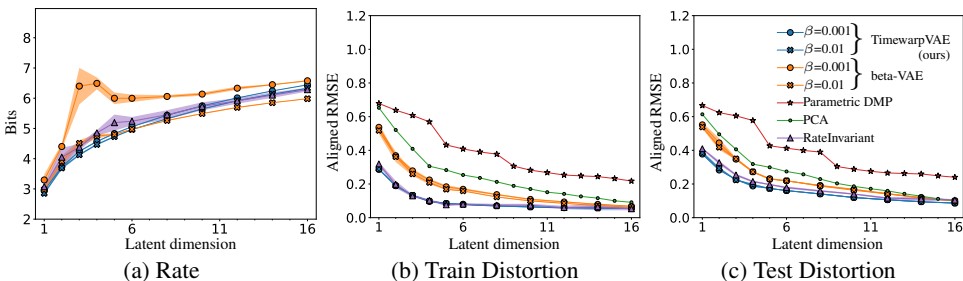

Figure 5: TimewarpVAE compared to beta-VAE, Parametric DMP (Matsubara et al., 2011), PCA, and Rate Invariant Autoencoder (Koneripalli et al., 2020). Especially for lower-dimensional models, TimewarpVAE shows comparable performance to RateInvariant on training error, and consistently underperforms RateInvariant in test error.

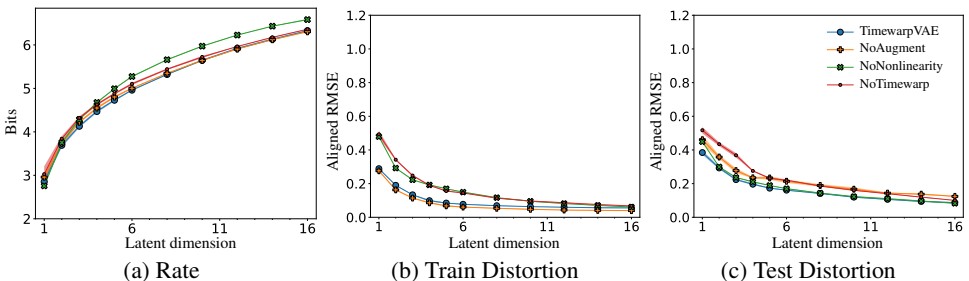

Figure 6: Ablation results show that data augmentation through timing noise is important for generalization performance, that our non-linear model gives a better fit to training data without losing generalization performance, and that the time-warper is key to TimewarpVAE's good performance.

generalization. However, NoNonlineary has such strong regularization that it is not able to get as good a fit to training data. Additionally, the model rate for NoNonlinearity is higher, showing that by constraining our model to be linear in the latent term, we are not able to compress information as efficiently into the latent space. Without the ability to time align the data, NoTimewarping is not able to fit either the training or test data well. The information rate of this model is the same as that of TimewarpVAE, showing that it does not compress spatial information as efficiently.

## 6 DISCUSSION

TimewarpVAE is useful for simultaneously learning timing variations and latent factors of spatial variations. Because it separately models timing variations, TimewarpVAE concentrates the modeling capacity of its latent variable on spatial factors of variation. As discussed further in Appendix A.2.1, TimewarpVAE can be viewed as a natural extension of Continuous DTW. We use a piecewise-linear, parametric model for the time-warping function. Another approach, might be to use a non-parametric model like DTW to warp the reconstructed trajectory to the input trajectory and then backpropagate the aligned error. That alternate approach has a much higher computation cost, because it requires computing DTW for each trajectory at each learning step, and does not re-use the time-warper from previous steps. We consider this approach briefly in Appendix A.12, and leave it to future work to more fully understand the benefits of this alternate implementation, and when it should be used or combined with the proposed TimewarpVAE. This work measured the spatial error of reconstructed training and test trajectories, and showed the TrajectoryVAE does a better job than beta-VAE at compressing spatial information into small latent spaces. Future work will investigate the tradeoff from compressing the timing information into smaller latent spaces.

REPRODUCIBILITY STATEMENT

Our experiments are performed on a publicly available human gesture dataset of air-handwriting (Chen et al., 2012), and on a dataset of quasistatic manipulation which we make publically available at `https://github.com/anonymousauthor913/iclr2024submission`. The PyTorch implementation of TimewarpVAE used in our experiments is also included at that url, as well as the code to generate the figures for this paper.

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

# A  APPENDIX

## A.1  BETA-VAE LOSS FUNCTION

The beta-VAE architecture is shown in Fig 7. Its loss function is $\mathcal{L} = \mathcal{L}_R + \mathcal{L}_{\mathrm{KL}}$, where

$$\mathcal{L}_R = \frac{1}{\sigma_R^2} \mathbb{E}_{x_i, t, \epsilon} \left[ \| x_i(t) - f\left(e(x_i) + \epsilon\right) \|^2 \right] \tag{6}$$

$$\mathcal{L}_{\mathrm{KL}} = \beta \mathbb{E}_{x_i} \left[ \frac{\| e(x_i) \|^2 + \sum_d (\sigma_d^2(x_i) + \log(\sigma_d^2(x_i)))}{2} \right] \tag{7}$$

$\mathcal{L}_R$ is a reconstruction loss that encourages the model to encode important information about the trajectories, and $\mathcal{L}_{\mathrm{KL}}$ acts as a rate regularization, constraining the total amount of information that the model is able to store about the trajectory (Alemi et al., 2017). For clarity, we use the subscript $x_i$, to emphasize that these losses are computed as empirical expectations over each of the training trajectories. $\sigma_d^2$ are the elements of $\sigma^2$. $\epsilon$ is pulled from a normal distribution with mean 0 and diagonal covariance given by $\sigma$. $\beta$ is a regularization hyperparameter for beta-VAEs.

## A.2  DERIVATION OF TIMEWARPVAE FROM DYNAMIC TIME WARPING

Dynamic Time Warping (DTW) compensates for timing differences between two trajectories by retiming the two trajectories so that they are spatially close to each other at matching timestamps. In this section, we explicitly derive TimewarpVAE from Continuous Dynamic Time Warping, the formulation of Dynamic Time Warping for continuous functions presented by Kruskal & Liberman (1983).

### A.2.1  CONTINUOUS DYNAMIC TIME WARPING

We begin with a brief summary of Continuous DTW. Given two trajectories $x_0$ and $x_1$ (each a function from time in $[0, 1]$ to some position in $\mathbb{R}^n$), Continuous DTW learns two time-warping functions, $\rho_0$ and $\rho_1$, where each time-warping function is monotonic and bijective from $[0, 1]$ to $[0, 1]$. The goal is to have the time-warped trajectories be near each other at corresponding (warped) timesteps. Mathematically, $\rho_0$ and $\rho_1$ are chosen to minimize the integral

$$\int_0^1 \| x_1\left(\rho_1(s)\right) - x_0\left(\rho_0(s)\right) \|^2 \, \frac{(\rho_0)'(s) + (\rho_1)'(s)}{2} \, ds \tag{8}$$

This integral is the distance between the trajectories at corresponding timesteps, integrated over a symmetric weighting factor.

This algorithm has a degeneracy, in that many different $\rho_0$ and $\rho_1$ will lead to equivalent alignments $\rho_1 \circ \rho_0^{-1}$ and will therefore have equal values of our cost function. This becomes relevant when generating new trajectories from the interpolated model, as it requires choosing a timing for the generated trajectory.

### A.2.2  REFORMULATION OF CONTINUOUS DTW

The optimization criterion of Continuous DTW can be rewritten as follows: Given the time-warping functions $\rho_0$ and $\rho_1$ from above, define $\phi_0 = \rho_0^{-1}$ and $\phi_1 = \rho_1^{-1}$. Let the function $f : [0, 1] \times [0, 1] \to \mathbb{R}^n$ be defined as $f(s, z) = (1 - z)x_0(\rho_0(s)) + zx_1(\rho_1(t))$. That is, $f(s, z)$ is the unique function that is linear in its second parameter and which satisfies the boundary conditions $f(\phi_0(t), 0) = x_0(t)$ and $f(\phi_1(t), 1) = x_1(t)$. These boundary conditions associate $x_0$ with $z_0 = 0$ and $x_1$ with $z_1 = 1$.

Our minimization objective for choosing $\rho_0$ and $\rho_1$ (or, equivalently, for choosing their inverses $\phi_0$ and $\phi_1$) can be written in terms of $f$ as

$$\frac{1}{2} \int_0^1 \left\| \frac{\partial f(s, z)}{\partial z} \bigg|_{s = \phi_0(t), z = 0} \right\|^2 dt + \frac{1}{2} \int_0^1 \left\| \frac{\partial f(s, z)}{\partial z} \bigg|_{s = \phi_1(t), z = 1} \right\|^2 dt. \tag{9}$$

The derivation goes as follows: We define $\phi_0 = \rho^{-1}$ and $\phi_1 = \rho^{-1}$, and we define the function $f : [0, 1] \times [0, 1] \to \mathbb{R}^n$ to be the unique function that is linear in its second parameter and which

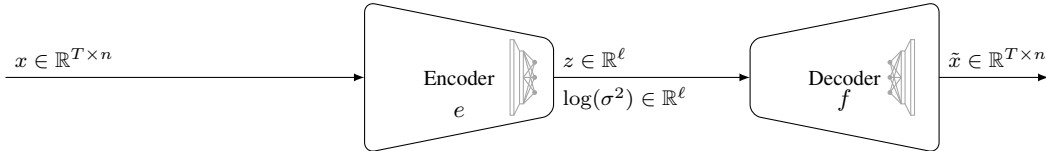

Figure 7: The architecture for Beta-VAE. Beta-VAE takes in a trajectory $x$, encodes it into a latent distribution parameterized by $z$ and $\log(\sigma^2)$, and decodes to a trajectory $\tilde{x}$.

satisfies the boundary conditions $f(\phi_0(t), 0) = x_0(t)$ and $f(\phi_1(t), 1) = x_1(t)$. Equivalently, it satisfies the boundary conditions $f(s, 0) = x_0(\phi_0^{-1}(s))$ and $f(s, 1) = x_1(\phi_1^{-1}(s))$.

Substituting these definitions gives an optimization criterion of

$$\int_0^1 \left\| x_1\left(\phi_1^{-1}(s)\right) - x_0\left(\phi_0^{-1}(s)\right) \right\|^2 \frac{\left(\phi_0^{-1}\right)'(s) + \left(\phi_1^{-1}\right)'(s)}{2} \, ds \tag{10}$$

The boundary conditions of $f$ imply this is equal to

$$\frac{1}{2} \int_0^1 \|f(s,1) - f(s,0)\| \, d\phi_0^{-1}(s) + \frac{1}{2} \int_0^1 \|f(s,1) - f(s,0)\| \, d\phi_1^{-1}(s) \tag{11}$$

And now, performing a change-of-variables $u = \phi_0^{-1}(s)$ and $v = \phi_1^{-1}(s)$ gives

$$\frac{1}{2} \int_0^1 \|f(\phi_0(u),1) - f(\phi_0(u),0)\|^2 \, du + \frac{1}{2} \int_0^1 \|f(\phi_1(v),1) - f(\phi_1(v),0)\|^2 \, dv \tag{12}$$

Since $f$ is linear in its second coordinate, we can write this in terms of the partial derivatives of $f$

$$\frac{1}{2} \int_0^1 \left\| \frac{\partial f(s,z)}{\partial z} \Big|_{s=\phi_0(u),z=0} \right\|^2 \, du + \frac{1}{2} \int_0^1 \left\| \frac{\partial f(s,z)}{\partial z} \Big|_{s=\phi_1(v),z=1} \right\|^2 \, dv \tag{13}$$

### A.2.3 SIMULTANEOUS TIME-WARPING AND MANIFOLD-LEARNING ON TRAJECTORIES

The relation to TimewarpVAE is as follows:

For each trajectory $x_i$, TimewarpVAE learns a low-dimensional latent representation $z_i \in \mathbb{R}^\ell$ associated with that trajectory. These $z_i$ are the natural generalizations of the reformulation of Continuous DTW above, which had hard-coded latent values $z_0 = 0$ and $z_1 = 1$ for the two trajectories.

For each trajectory $x_i$, TimewarpVAE learns a time warping function $\phi_i$ that transforms timesteps to new canonical timings. These $\phi_i$ are the natural extension of the $\phi_0$ and $\phi_1$ from above.

TimewarpVAE learns a generative function $f$ which, given a canonical timestamp $s$ and a latent value $z$, returns the positon corresponding to the trajectory at that time. This is an extension of the function $f$, with relaxations on the linearity constraint and the boundary conditions. Instead of requiring $f$ to be linear in the $z$ argument, we parameterize $f$ with a neural network and regularize it to encourage $f$ to have small partial derivative with respect to the latent variable $z$. This regularization is described in Section A.2.4. Instead of a boundary constraints requiring $f(\phi_i(t), z_i)$ to be equal to $x_i(t)$, we instead add an optimization objective that $f(\phi_i(t), z_i)$ be close to $x_i(t)$,

### A.2.4 REGULARIZATION OF THE DECODER

Training the decoder using our optimization objective (Eq. 1) includes adding noise $\epsilon$ to the latent values $z$ before decoding. This encourages the decoder to take on similar values for nearby values of $z$. In particular, as described by Kumar & Poole (2020), this will add an implicit Jacobian squared regularization of the decoder over the $z$ directions. Penalizing these $\|\frac{\partial f(s,z)}{\partial z}\|^2$ terms is exactly what we want for our manifold-learning algorithm. Additionally, we note that we do not add any noise to the temporal encoder when computing the reconstruction loss, so our beta-VAE style architecture does not include any unwanted regularization of $\|\frac{\partial f(s,z)}{\partial s}\|^2$.

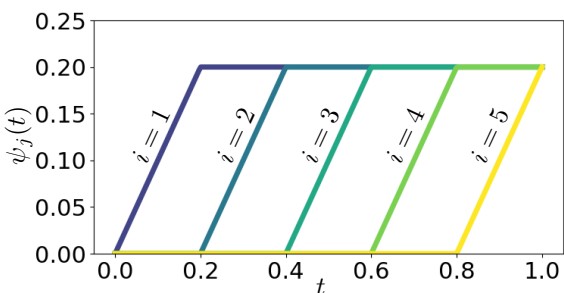

Figure 8: Time alignment basis functions for $K = 5$, for each $i$ from 1 to 5

### A.3 DEGENERACY OF TIME WARPING USING NOTATION OF SHAPIRA WEBER ET AL. (2019)

Using the notation of Shapira Weber et al. (2019), the degeneracy noted in Sec. 4.5 also applies. For any set of time warping functions $T^{\theta_i}$, one for each of the $N_k$ different trajectories $u_i$ with class label $y_i$ out of $K$ possible class labels, you can compose all of those time warping functions with some additional fixed diffeomorphic warping $T^p$ without changing the Inverse Consistency Averaging Error.

That is, composing all time warps with a time warping function $T^p$ to generate $\tilde{T}^{\theta_i} = T^{\theta_i} \circ T^p$ will not affect the Inverse Consistency Averaging Error, since now the (perturbed) average warped trajectory for cluster $k$ $\tilde{\mu}_k$ is the warp of the previous average warped trajectory $\mu_k$

$$\tilde{\mu}_k = \frac{1}{N_k} \sum u_i \circ T^{\theta_i} \circ T^p = \frac{1}{N_k} \left( \sum u_i \circ T^{\theta_i} \right) \circ T^p = \mu \circ T^p \tag{14}$$

.

The inverse $\tilde{T}^{-\theta_i}$ is simply $T^{-p} \circ T^{-\theta_i}$

The Inverse Consistency Averaging Error using those perturbed time warps $\tilde{T}$ is the same as that computed using the original time warps.

$$\mathcal{L}_{ICAE}(\tilde{T}) = \sum_{k=1}^{K} \frac{1}{N_K} \sum_{i:y_i=k} \left\| \tilde{\mu}_k \circ \tilde{T}^{\theta_i} - u_i \right\|_{\ell_2}^2 \tag{15}$$

$$= \sum_{k=1}^{K} \frac{1}{N_K} \sum_{i:y_i=k} \left\| \mu_k \circ T^p \circ T^{-p} \circ T^{-\theta_i} - u_i \right\|_{\ell_2}^2 \tag{16}$$

$$= \sum_{k=1}^{K} \frac{1}{N_K} \sum_{i:y_i=k} \left\| \mu_k \circ T^{-\theta_i} - u_i \right\|_{\ell_2}^2 \tag{17}$$

$$= \mathcal{L}_{ICAE}(T) \tag{18}$$

This shows that there is a degeneracy over the choice of warping of the warped trajectories. Warped trajectories can all be additionally warped by another time warping function without changing the Inverse Consistency Averaging Error.

### A.4 PLOT OF TIME-WARPING BASIS FUNCTION

A plot of the time-warping basis functions $\psi$ is shown in Fig. 8.

### A.5 EXAMPLE TRAINING TRAJECTORIES OF LETTER "A"

Example training trajectories of handwritten letter "A"'s are shown in Fig. 9.

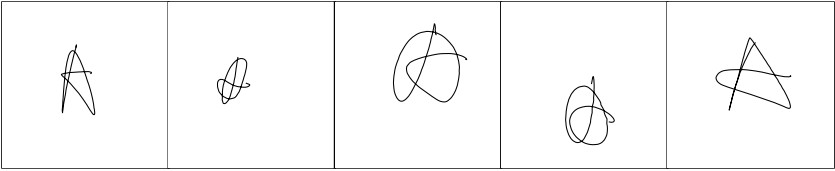

Figure 9: Five example trajectories of handwritten "A"

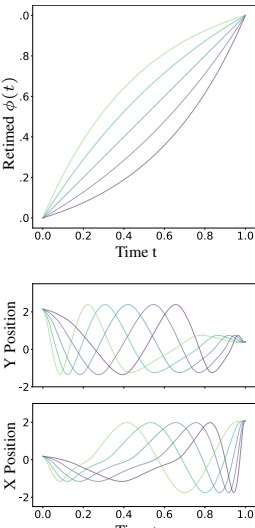

Figure 10: Decoding the same spatial latent variable using different timewarping parameters will give the same spatial trajectory but with different timings (fast or slow at different times). We plot the time warping functions for five different timing latents and the generated trajectories for a single spatial latent value by showing the generated positions as a function of time.

### A.6 TIME WARPER EFFECT ON TRAINED MODEL

In Fig. 10 we show how varying the timewarper parameters affects the timing of the generated trajectory for one of the TimewarpVAE letter models. Here, we choose five different sets of timewarper parameters, and plot the resulting timewarper function $\phi(t)$. We then decode the same spatial trajectory using those five latent parameters. The resulting trajectories spatially would look all the same, and we plot the X and Y locations of the generated trajectories as a function of time. The associated timings of the trajectories changes due to the timewarper function.

### A.7 ADDITIONAL INTERPOLATIONS FOR MODELS

Here, we present additional interpolation results, all on 16 latent dimensions and $\beta = 0.001$. We note that the convolutional encoder/decoder architecture in beta-VAE in Fig. 11b does not appear to have as strong an implicit bias toward smooth trajectories as the TimewarpVAE architecture. This makes sense because the TimewarpVAE architecture decomposes the generative function into a component $g(s)$ which computes poses as a function of time, likely inducing an inductive bias toward smoother trajectories as a function of time.

The interpolation in Fig. 11a shows that in the ablation of TimewarpVAE without the timing module the interpolation does not preserve the style of the "A". Likewise, the DMP interpolation in Fig. 11c does not preserve the style of the "A".

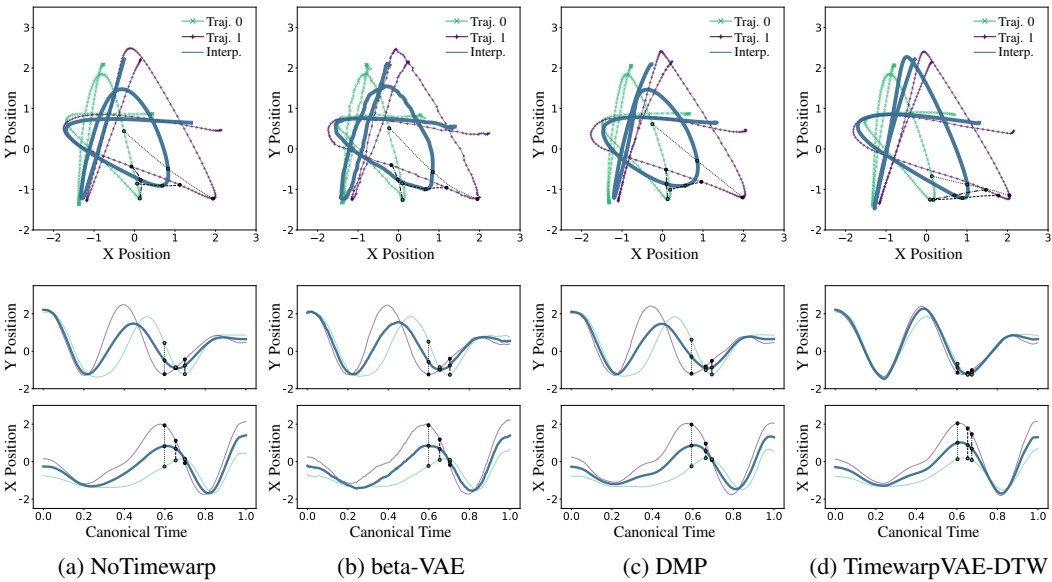

|  |  |  |  |
|---|---|---|---|
| (a) NoTimewarp | (b) beta-VAE | (c) DMP | (d) TimewarpVAE-DTW |

Figure 11: Additional interpolation results

## A.8 EQUIVALENCE OF REGULARIZING $\phi$ OR $\phi^{-1}$

We note that our regularization is the same, regardless if we regularize $\phi$ or $\phi^{-1}$. We show this by applying the substitution $s = \phi(t)$, $ds = \phi'(t)dt$. That substitution gives: $\int_0^1 \left(1 - \frac{1}{\phi'(\phi^{-1}(s))}\right) \log\left(\phi'(\phi^{-1}(s))\right) ds$.

We use the identity $(\phi^{-1})'(s) = \frac{1}{\phi'(\phi^{-1}(s))}$ to simplify to

$$\int_0^1 \left(1 - (\phi^{-1})'(s)\right) \log\left(\frac{1}{(\phi^{-1})'(s)}\right) ds = \int_0^1 \left((\phi^{-1})'(s) - 1\right) \log\left((\phi^{-1})'(s)\right) ds \qquad (19)$$

This is exactly our regularization applied to the function $\phi^{-1}$. Thus, we note that our regularization is symmetric. Our regularization cost is the same whether it is applied to $\phi$ (the function from trajectory time to canonical time) or applied to $\phi^{-1}$ (the function from canonical time to trajectory time).

It didn't have to be that way. For example, if our cost function were of the form $\int_0^1 \log^2(\phi'(t))dt$, the substitution above would give a cost $\int_0^1 \frac{1}{\phi'(\phi^{-1}(s))} \log^2(\phi'(\phi^{-1}(s)))ds$ which simplifies to $\int_0^1 \frac{1}{\phi'(\phi^{-1}(s))} \log^2((\phi^{-1})'(s))ds$ which equals $\int_0^1 (\phi^{-1})'(s) \log^2((\phi^{-1})'(s))ds$ which is different from applying the regularization procedure to $\phi^{-1}$ which would have given a regularization term of: $\int_0^1 \log^2((\phi^{-1})'(s))ds$

## A.9 INSPIRATION FOR REGULARIZATION FUNCTION

Our regularization function was inspired by Unbalanced Optimal Control. If we assign a uniform measure $\mathcal{U}$ to $[0, 1]$, we note that our regularization cost is exactly the symmetric KL Divergenge between $\mathcal{U}$ and the pushforward $\phi_{i*}\mathcal{U}$. For each $\phi_i$, the pushforward $\phi_{i*}\mathcal{U}$ has a probability density function $1/(\phi_i'(\phi_i^{-1}(s)))$, and the symmetric KL divergence cost $D_{\mathrm{KL}}(\mathcal{U}|\phi_{i*}\mathcal{U}) + D_{\mathrm{KL}}(\phi_{i*}\mathcal{U}|\mathcal{U})$ gives the loss given above.

We work through the explicit mathematics below.

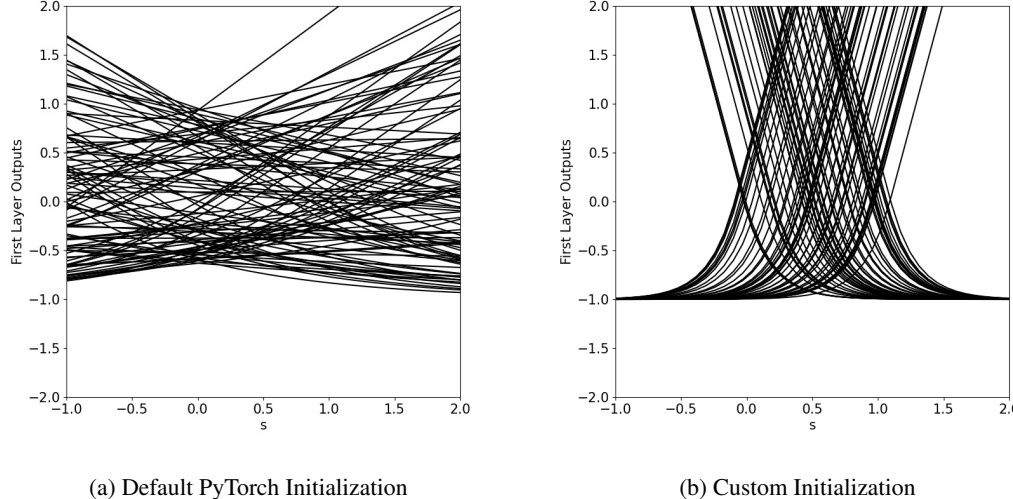

(a) Default PyTorch Initialization

(b) Custom Initialization

Figure 12: Outputs of first layer of the neural network $g(s)$.

### A.9.1 PUSHFORWARD OF PROBABILITY DENSITY FUNCTION

If $F_0$ is some cumulative distribution function, and $F_1$ is the CDF generated by the pushforward of a function $\phi$ then we have the simple identity $F_1(\phi(t)) = F_0(t)$. Taking the derivative of both sides with respect to $t$ gives $F_1'(\phi(t))\phi'(t) = F_0'(t)$. The substitutions $s = \phi(t)$ and $\phi^{-1}(s) = t$ give $F_1'(s) = \frac{1}{\phi'(\phi^{-1}(s))}F_0'(\phi^{-1}(s))$. Since the PDF is the derivative of the CDF, then writing $f_0$ and $f_1$ as the corresponding PDFs of $F_0$ and $F_1$, we see

$$f_1(s) = \frac{1}{\phi'(\phi^{-1}(s))} f_0(\phi^{-1}(s)) \tag{20}$$

In the simple case where $f_0 = \mathcal{U}$, the uniform distribution over $[0, 1]$, then $f_0(t) = 1$, so we have our pushforward

$$(\phi_*\mathcal{U})(s) = \frac{1}{\phi'(\phi^{-1}(s))} \tag{21}$$

### A.9.2 SYMMETRIC KL DIVERGENCE CALCULATION

The KL divergence $D_{\mathrm{KL}}(\mathcal{U}|\phi_*\mathcal{U})$ is $\int_0^1 \log\left(\phi'(\phi^{-1}(s))\right) ds$. Substituting $\phi(t) = s$ and the corresponding $\phi'(t)dt = ds$ gives $\int_0^1 \phi'(t) \log\left(\phi'(t)\right) dt$. Likewise, the KL divergence $D_{\mathrm{KL}}(\phi_*\mathcal{U}|\mathcal{U})$ is $\int_0^1 \frac{1}{\phi'(\phi^{-1}(s))} \log\left(\frac{1}{\phi'(\phi^{-1}(s))}\right) ds$, which simplifies with the same substitutions to $-\int_0^1 \log\left(\phi'(t)\right) dt$.

The symmetric KL divergence cost is thus

$$D_{\mathrm{KL}}(\mathcal{U}|\phi_*\mathcal{U}) + D_{\mathrm{KL}}(\phi_*\mathcal{U}|\mathcal{U}) = \int_0^1 (\phi'(t) - 1) \log\left(\phi'(t)\right) dt \tag{22}$$

### A.10 INITIALIZATION OF NEURAL NETWORK FOR $f(s, z)$

As mentioned above, the neural network $f(s, z)$ is split into $\mathbf{T}(z)$ and $g(s)$. Since $g(s)$ takes in a canonical time $s \in [0, 1]$ which we want to have roughly uniform modeling capacity over the full range of $s$ from 0 and 1, we initialize the first layer of the neural network's weights, $W$ (a matrix with one column), and bias $b$ (a vector) for $g(s)$ in the following way.

We initialize the values in $W$ to be randomly, independently $-G$ or $G$ with equal probability, where $G$ is a hyperparameter.

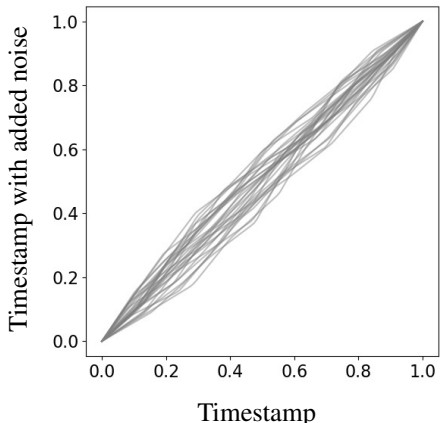

Figure 13: Example (random) functions used to add timing noise during data augmentation

We then choose values of $b$ so that, for each (output) row $j$, which we denote $W_j$ and $b_j$ the y-intercepts of the function $y = g_j(s) = W_j s + b_j$ are each an independent uniformly random value in $[0 - \eta, 1 + \eta]$

Visually, the effect of this initialization can be seen by plotting the first layer's transformation $\text{ELU}(Ws + b)$ using PyTorch's default initialization and using our proposed initialization, where ELU is the Exponential Linear Unit introduced by Clevert et al. (2016). PyTorch's default implementation randomly initializes the output functions to be distributed symmetrically around $s = 0$. Additionally, much of the modeling capacity is assigned to variations outside the domain $[0, 1]$. Since we know that the input timestamp will be $s \in [0, 1]$, our initialization focuses the modeling capacity near $[0, 1]$ and is symmetric around the middle of that range $s = 0.5$.

## A.11 TIMING NOISE DATA AUGMENTATION

For data augmentation of timing noise, we create perturbed timesteps to sample the training trajectories as follows. First, we construct two random vectors $\nu_{\text{in}}$ and $\nu_{\text{out}}$ of uniform random numbers between 0 and 1, each vector of length 10. We then square each of the elements in those vectors and multiply by a noise hyperparameter $\eta$ and take the cumulative sum to give perturbation vectors for input and output timings. We add each of those vectors to a vector with ten elements with linear spacing, $[0, 1/9, 2/9, 3/9, \ldots, 1]$. We then normalize those perturbed vectors so that the last elements are again 1. The resulting vectors now give nicely symmetric, monotonic x and y coordinates of knots for a stepwise-linear perturbation vector which we can subsample at arbitrary timesteps to give desired noise-added output timesteps. We choose $\eta = 0.1$ in our experiments, giving noise functions plotted in Fig. 13.

When using this data augmentation, each time we pass a training trajectory into our model, instead of sampling the training trajectory at $T$ uniformly-distributed timesteps between 0 and 1 to construct our $x \in \mathbb{R}^{T \times n}$, we first perturb all those timesteps by passing them through a randomly generated noise functions. This means that each $x$ we pass into our learning algorithm has a slightly different timing than the training data, allowing us to perform data augmentation on the timings of the training data.

## A.12 ALTERNATIVE TIMEWARPVAE APPROACH DIRECTLY USING DYNAMIC TIME WARPING

An alternative formulation of TimewarpVAE, which we call TimewarpVAE-DTW, is to collect the decoded trajectory as a vector by running the decoder $f(s, z)$ over multiple, evenly-sampled canonical timesteps $s$, and then warping those generated timesteps to the training data using DTW. This is more similar to Rate Invariant Autoencoder, in that the time warping happens after trajectory

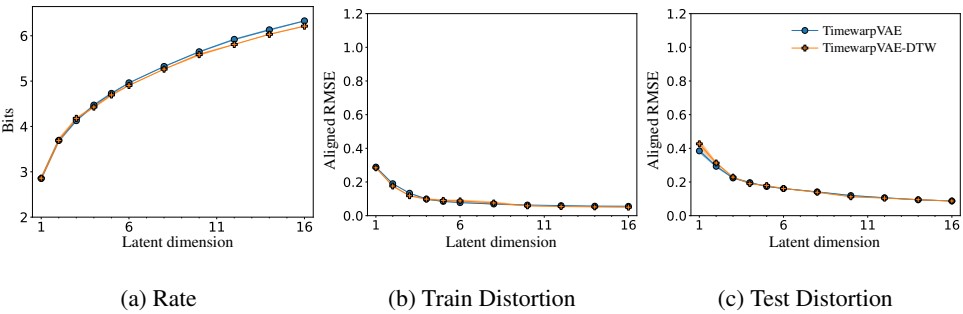

(a) Rate           (b) Train Distortion          (c) Test Distortion

Figure 14: Performance comparison between TimewarpVAE and TimewarpVAE-DTW

generation, however, we do not require linear interpolation. Instead, we convert the DTW alignment into a loss in such a way that all canonical trajectory points are used and averaged (using the same weightings we descriped for our Aligned RMSE). This avoids the problem encountered in Rate Invariant Autoencoder where parts of the canonical trajectory can be completely ignored. TimewarpVAE-DTW requires running DTW to align each reconstructed trajectory to its associated training trajectory every time the decoder function is executed during training. This is significantly less efficient than our suggested implementation of TimewarpVAE, because it requires many executions of Dynamic Time Warping with no re-use of the DTW results between training steps. Our suggested implementation, TimewarpVAE, explicitly models the time-warping, so is able to re-use (and update) the warping function between steps, rather than recalculating it from scratch each time. However, we note in Fig.14 that TimewarpVAE-DTW, though less efficient, can give comparable results. In this implementation we use DTW, but a Soft-DTW (Cuturi & Blondel, 2017) could be used instead.

### A.13    HYPERPARAMETERS

The specific training architectures we use is shown in Table 2. We use a kernel size of 3 for all convolutions. $e$ is the spatial encoding architecture (which is always reshaped to a vector and followed by two separate fully-connected layers, one outputting the expected encoding, and one outputting the log of the diagonal of the covariance noise). $h$ is the temporal encoding architecture, which is always followed by a fully-connected layer outputting a vector of size $K = 50$. $g(s)$ is part of the factorized decoder architecture, which is followed by a fully-connected layer outputing a vector of size $m = 64$. $\mathbf{T}(z)$ is the other part of the factorized decoder architecture, which is followed by a fully-connected layer outputting a vector of size $nm$ where $n$ is the number of channels in the training data (2 for handwriting, 7 for fork trajectories) and $m = 64$. For beta-VAE, instead of the factorized decoder architecture, we use one fully-connected layer with output size $800$, which we then reshape to size $25 \times 32$. This is followed by one-dimensional convolutions. Following the approach of Kuester et al. (2021), for the convolutions in the beta-VAE architecture, instead of doing convolutional transposes with strides to upsample the data, we instead always use a stride length of 1 and upsample the data by duplicating each element before performing each convolution. Thus, the lengths expand from 25 in the input to the output size of 200 after the three convolutions.

We use a learning rate of 0.0001, a batch size of 64, and a rectified linear unit (ReLU) for all spatial and temporal encoder nonlinearities except for Rate Invariant Autoencoder for which follow the literature and we Tanh, and we use an exponential linear unit (ELU) for the decoder nonlinearities for TimewarpVAE, we use ReLU for the decoder nonlinearities for beta-VAE, and we again use Tanh for the Rate Invariant Autoencoder. We choose a variance estimate of $\sigma_R^2 = 0.01$ for our data, but this hyperparameter is not critical, as it is equivalent to scaling $\beta$ and $\lambda$ in Eq. 1. In order to compute the Rate (information bottleneck) of the Rate Invariant Autoencoder, we implement it as a VAE instead of an autoencoder, only adding noise to the spatial latent, not to the timing latent values. We use 199 latent variables for the timing (one fewer than the trajectory length), and vary the number of spatial latent variables.

Table 2: Hyperparameters

| Name | $e$ conv channels strides | $h$ conv channels strides | $g(s)$ fc | $\mathbf{T}(z)$ fc | $f$ fc conv channels |
|---|---|---|---|---|---|
| TimewarpVAE | [16,32,64,32] [1,2,2,2] | [16,32,32,64,64,64] [1,2,1,2,1,2] | [500,500] | [200] | – |
| NoTimewarp | [16,32,64,32] [1,2,2,2] | – | [500,500] | [200] | – |
| NoNonlinearity | [16,32,64,32] [1,2,2,2] | [16,32,32,64,64,64] [1,2,1,2,1,2] | [500,500] | [] | – |
| beta-VAE | [16,32,64,32] [1,2,2,2] | – | – | – | [800] [20,20,$n$] |
| Rate Invariant Autoencoder | [32,32,32] [1,1,1] | – | – | – | [6400] [32,32,$n$] |

