# OpenReview forum: "TimewarpVAE: Simultaneous Time-Warping and Representation Learning of Trajectories"
_ICLR.cc/2024/Conference — Submitted to ICLR 2024_

### Official Review · Reviewer_joWW · 2023-10-29

**Soundness:** 3 good
**Presentation:** 4 excellent
**Contribution:** 3 good
**Rating:** 6
**Confidence:** 3

**Summary:**

The paper proposes Timewarp VAE, which simultaneously learns both spatial variations with temporal variations. It is based on beta-VAE with two additional modules: a temporal encoder and a time-warper, the decoder then takes the canonical trajectory and time, which are trained jointly to enable good reconstruction of position trajectory at each timestep. Experimental results on small handwriting and fork manipulation datasets show superior performance compared with baseline approaches.

**Strengths:**

The paper is very well written.
The paper proposes a neat idea to incorporate temporal information into existing beta-VAE methods.
The paper has a range of ablation studies in the empirical study to examine the chosen architecture.

**Weaknesses:**

There is limited comparison with other approaches such as sequence-to-sequence type of architecture. The decoder is a sequence of fully connected layers, if using some richer architecture like RNN/transformer type of models which can incorporate temporal information, it is not clear how much value the proposed time warper adds. The paper shows results on two experiments: collected fork manipulation data and a small handwringing gestures dataset. More experimental on other datasets will also make the paper stronger.

**Questions:**

The ablation study on ‘NoTimewarp’ studies the performance when replacing the time-warper with identity function, which had inferior performance indicating the importance of the time warper. I wonder if the input trajectories are represented as equi-spaced sequences (i.e., with some temporal information), where good reconstruction would include both the spatial reconstruction and temporal reconstruction, how does the model perform.

Do all the training trajectories have the same length T?

What is the intuition of lambda, e.g., what does it mean when having small lambda vs big lambda?

These papers may be related: https://www.sciencedirect.com/science/article/pii/S0925231220312017
https://openreview.net/pdf?id=Byx1VnR9K7

---

> ### Author Response · Authors · 2023-11-18
>
> ## Introduction
> The authors would like to thank reviewer joWW for this review. We have added the reviewer's suggested related works "TrajVAE: A Variational AutoEncoder model for trajectory generation" and "Trajectory VAE for Multi-Modal Imitation" to our Related Work section and explain how our work is different from those works. Those works incorporate timing and spatial information together into a combined latent space, while our approach is able to learn a separate embedding of spatial and timing vectors. Additionally, we have improved the manuscript now to clarify the questions mentioned by the reviewer, that the trajectories are all sampled to length T=200, and to provide some intuition for the regularization term $\lambda$.
>
> ## Is it possible to pass in the input trajectories as sequences (with some temporal information appended alongside them) like you would for a Transformer or RNN?
> The TrajectoryVAE formulation of an encoder, decoder, and timewarper could easily accommodate a variety of different neural network architectures, including RNNs or Transformers. We would still be able to separate the latent space into timing and spatial components in the same way. The key is to have two separate encoders (the spatial encoder and the timing encoder), and then the decoder would generate the sequence using the spatial latent vector and using timestamps that have been warped by the time warper. We have tried a Transformer decoder architecture, for example, with similar results, but the model training time was too long to do an exhaustive analysis.
>
>  ## If good reconstruction would include both the spatial reconstruction and temporal reconstruction, how does the model perform?
> This is an interesting question. One downside to our model is that in the current implementation, we use a very large latent space for the time warping latent variables. This is because our focus was on compressing the spatial latent space to lead to good spatial reconstructions. We are able to fairly compare to other models with small latent spaces because we do not use the temporal information at all in our TimewarpVAE evaluations to generate the reconstructed trajectories and to compute our spatial reconstruction error.
>
> However, to do a fair comparison to measure both spatial reconstruction and temporal reconstruction between our model and other models which combine together the timing and spatial latent variables into a single latent space (like the TrajVAE and Trajectory VAE for Multi-Modal Imitation), we'd need to count the size of the timing latent space as well in order to be fair. In future work we plan to reduce the size of our timing latent space into a more compressed space (like we already do for the spatial information). Our intuition is that this will lead to an interpretable small latent space with separate variables for timing and separate variables for spatial information. However, our current approach is not designed to address this particular problem of "How small can the sum of the sizes of the timing latent space and the spatial latent space be?" Given the nuances of this question, we leave it to future work.
>
> ## Do all the training trajectories have the same length T?
> Yes, we have now added this clarification to the paper.
>
> ## What is the intuition of lambda, e.g., what does it mean when having small lambda vs big lambda?
> We have now added more insight to lambda in the paper. Since $\lambda$ penalizes the timewarping in the model, a large $\lambda \to \infty$ would drive time-warping to be the identity function (meaning the spatial latent space would have to dedicate some of its modeling capacity to also account for timing differences between trajectories), and $\lambda=0$ could allow the model to learn severe time warps (which is a problem we see in some models, where the time warping module learns to completely skip over parts of the template trajectories).
>
> ## These papers may be related "TrajVAE: A Variational AutoEncoder model for trajectory generation" and "Trajectory VAE for Multi-Modal Imitation"
> We have now added these papers to our related work section. The fundamental distinction between those papers and ours lies in their utilization of a combined latent space encompassing both timing and spatial variations, whereas our TimewarpVAE approach separates information about timing and spatial information into different latent spaces.
>
> ## Conclusion
> We again thank this reviewer for their questions, which have allowed us to modify the paper to clarify some details we previously missed and to strengthen our related work section.

---

> > ### Comment · Reviewer_joWW · 2023-11-22
> >
> > Thank you for addressing my concerns and the additional clarification in the paper, I appreciate the authors' efforts in the additional experiments and comparison with other methods which have greatly improved the paper. I would like to see it published.

---

### Official Review · Reviewer_qQMc · 2023-10-29

**Soundness:** 4 excellent
**Presentation:** 4 excellent
**Contribution:** 2 fair
**Rating:** 3
**Confidence:** 4

**Summary:**

The paper proposes a neural network architecture for time warp-invariant representation learning i.e., disentangle timing variations from the spatial variations in a dataset. The network contains separate modules for spatial and temporal encoding and a time warping module that maps the temporal encoding and an input timestamp to the output resampled timestamp. The spatial encoding, which is a beta-VAE, and the resampled timestamp are input to the decoder to obtain the input time series at that timestamp. All the modules trained end-to-end using the beta-VAE objective as well as a time-warping regularization term.

**Strengths:**

1. The paper is well-written. The ideas are also simple, intuitive and easy to implement.

2. Experiments show clear improvements over baseline beta-VAE.

**Weaknesses:**

I believe this paper has some fundamental weaknesses related to how novel the ideas are.

There are other works that I list here that are very close to this submission that make the ideas in the paper not very novel, in my opinion

(a) Diffeomorphic Temporal Alignment Nets: https://proceedings.neurips.cc/paper_files/paper/2019/file/db98dc0dbafde48e8f74c0de001d35e4-Paper.pdf
(b) SrvfRegNet: Elastic Function Registration Using Deep Neural Networks: https://openaccess.thecvf.com/content/CVPR2021W/DiffCVML/papers/Chen_SrvfRegNet_Elastic_Function_Registration_Using_Deep_Neural_Networks_CVPRW_2021_paper.pdf
(c) Rate-invariant autoencoding of time-series: https://ieeexplore.ieee.org/abstract/document/9053983
(d) Regularization-free Diffeomorphic Temporal Alignment Nets: https://openreview.net/pdf?id=7IbLWa0anE

Especially (c), it also has an encoder that produces a single rate-invariant latent vector for the full time series and the other part of the latent space is a time warp followed by a time warping module, and the network is trained using reconstruction loss.

These papers are already in literature and in light of them, the ideas in the paper do not appear to be very novel to me. It would be good if the authors can help me understand how this paper is different.

Even if the ideas are substantially different, some of these papers should appear as baselines in the experiments. At the moment, the only baseline is the beta-VAE, which predictably fails to capture the time-series misalignments.

**Questions:**

No additional questions.

---

> ### Author Response · Authors · 2023-11-18
>
> ## Introduction
> We thank reviewer qQMc for drawing our attention to these related works, especially the interesting paper "Rate-invariant autoencoding of time-series". We agree that this paper tackles the same problem addressed in our work, so we have updated our manuscript to clarify the differences in our approaches, we have added that method as a quantitative and qualitative baseline, and we have additionally provided some insight into why our approach is better at generating trajectories for execution on a robot. Additionally, we have now been able to provide an additional theoretical contribution by showing that the "Regularization-free Diffeomorphic Temporal Alignment Nets" did not properly consider the time warping degeneracy we address with our time warping regularization term.
>
> ## Rate-invariant autoencoding of time-series
> For "Rate-invariant autoencoding of time-series" (RIA), the primary differences are:
> 1) RIA is restricted to generating a canonical trajectory that is piecewise linear. RIA's canonical trajectory (the output of the convolutional neural network) gives the position at pre-specified time steps. Linear interpolation is used during warping to look up positions from the canonical trajectory. By contrast, our method can parameterize the template trajectory as an arbitrary neural network function of the canonical timestamp. This is especially important when combined with the second difference below.
> 2) The second difference is that RIA does not contain any regularization on the time warping function. This can allow the canonical trajectory to be arbitrarily warped compared to the data trajectories.
> 3) For completeness: additionally, we use a VAE instead of a simple autoencoder, and, related to point 1 above, our time warping mechanism happens before trajectory decoding, while in their approach it happens after trajectory decoding.
>
> Together, as we have updated our paper to show, these first two differences mean that RIA can easily learn to skip parts of the canonical trajectory during learning. That is, without the time warping regularization, it's easier for RIA to have severe warping, skipping large pieces of the canonical trajectory. Likewise, once RIA has learned to skip a portion of the canonical trajectory, the use of linear interpolation means that no gradient information will flow to the skipped parts of the canonical trajectory, so RIA will not learn to fix that part of the trajectory. Our approach does not have that issue. This comparison can now be seen qualitatively in Figure 1 of our updated paper and quantitatively in our updated results section.
>
> ## "Diffeomorphic Temporal Alignment Nets" and "Regularization-free Diffeomorphic Temporal Alignment Nets"
> Additionally, we would like to thank the reviewer for drawing our attention to the interesting papers "Diffeomorphic Temporal Alignment Nets" and "Regularization-free Diffeomorphic Temporal Alignment Nets". We found it especially interesting to consider the claim in the second paper that any regularization of time warping is not needed, since that argument presented there would also translate to our work. We find, however, that "Regularization-free Diffeomorphic Temporal Alignment Nets" does not properly consider the degeneracy of the overall choice of timing for the canonical trajectory (which we address in Section 4.5). To better align our contributions with the literature, we have additionally added Appendix A.3, in which we prove that this timing degeneracy also applies to the Inverse Consistency Averaging Error proposed in that work, a degeneracy which could be solved by adding a time warping regularization term like ours.
>
> ## "SrvfRegNet: Elastic Function Registration Using Deep Neural Networks"
> "SrvfRegNet: Elastic Function Registration Using Deep Neural Networks" gives another approach to properly handling the reconstruction loss when comparing warped trajectories (performing the type of change of variable analysis we perform in our Appendix A.2), we note that  "adding a constant to a function does not change its SRVF". Though we mean-center our overall training set of trajectories, individual trajectories will have different mean positions, and we want our model to be able to differentiate the spatial position of trajectories, as that is important when executing a trajectory on a robot.
>
> ## Conclusion
> The authors would again like to thank reviewer qQMc for this review, as it has significantly improved this manuscript, both in terms of better baseline comparisons of our method, and also allowing a clearer theoretical contribution on the importance of the time warper regularization term.

---

> > ### Comment · Reviewer_qQMc · 2023-11-22
> > **Thank you for the author response**
> >
> > Thank you for the clarification of novelty in context of existing works.
> >
> > In light of the modifications, the paper is better in my opinion.
> >
> > But the novelty is rather limited. The main difference seems to be using a neural network for interpolation rather than linear interpolation. Regularization functions for time warping are well-known in general to avoid degeneracy issues, also used Diffeomorphic Temporal Alignment Nets.
> >
> > Based on the above, I am inclined to maintain my score.

---

### Official Review · Reviewer_ui6D · 2023-10-31

**Soundness:** 2 fair
**Presentation:** 3 good
**Contribution:** 3 good
**Rating:** 6
**Confidence:** 4

**Summary:**

The paper proposes to explicitly learn a time-warping network in learning VAE-based trajectory distributions. The motivation lies in decomposing models that capture spatial and temporal variations and aligning a set of trajectories to a manifold with canonical time index, unike pair alignment in traditional Dynamic Time Warping. The paper also discusses several design choices to further improve the performance, including data augmentation based on time-perturbation, regularisation of the time-warping network output and linear/nonlinear basis in decoder networks. The proposed model is validated in handwriting in-air and fork manipulation datasets. The proposed TimewarpVAE outperforms baseline beta-VAE and ablative versions by reconstructing samples that can be better spatially aligned with the test input. Qualitative result also shows a preferred interpolation performance in the latent space on the handwriting dataset.

**Strengths:**

* The paper writing is clear and it is especially commendable on the detailed considerations and theoretical insights on the introduction of loss, regularisation terms, network design and association to continuous DTW formulations.

* The motivation of attempting to decouple the modelling of temporal and spatial variabilities is well grounded and can find many applications in modelling temporal data.

* The idea is easy to follow and seems to work with a few evidence on better model compression and generalisation capacity.

**Weaknesses:**

* The results could be more convincing if the experiments can go beyond low-dimensional data. Both handwriting and fork data are limited to gesture pose that only pertains to a handful degree-of-freedom. beta-VAE appears to catch up when a larger latent space is used so it is unclear if the advantage of TimewarpVAE can persist in face of more spatial complexity.

* It is hard to tell if time warping and spatial networks are actually extracting the expected information. The results in interpretable latent space look circumstantial since it is only about spatial dimensions and not fully disentangled. beta-VAE indeed cannot realise disentanglement solely based on reconstruction, but still the independence between the spatial and temporal components is not sufficient addressed.

**Questions:**

* Can TimewarpVAE show advantageous performance on larger-scale dataset such as mocap skeletal data involving many human limbs?

* Can we have more direct evidence showing the spatial latent variable and canonical time index can independently control the variations of trajectories along the expected dimensions.

* The fork dataset has part as the scaled quaternion while the error appears to be evaluated in the sense of Euclidean. Will this impact training and evaluation?

* Can TimewarpVAE work with multiple data modalities, e.g. handwriting trajectories for all alphabetical letters? Will it need prior other than isotropic Gaussian? Will the identification of time variation help us to have a more structured latent space to group each data modal?

* How TimewarpVAE is related to other generative models with dynamical latent space, such as VAE-DMP [a] ?

[a] Chen et al, Dynamic movement primitives in latent space of time-dependent variational autoencoders, Humanoids 2016

---

> ### Author Response · Authors · 2023-11-18
>
> ## Introduction
> The authors would like to thank reviewer ui6D for their comments. We have now added Figure 10 to explicitly show the effect of the timewarper parameters on the generated trajectory, we have clarified that we preprocessed the quaternion representations to ensure they are near each other in $R^4$, and we have now differentiated our work from "Dynamic movement primitives in latent space of time-dependent variational autoencoders" in our Related Work section.
>
> ## Can TimewarpVAE show advantageous performance on data involving many human limbs?
>
> We do not think our model would necessarily suffer from high dimensional input spaces. The encoder already takes in a very high-dimensional input (on the size of 200 timesteps times 7 DOF = 1400 dimensions). There might need to be some optimization of the decoder network architecture to generate high dimensional outputs. To the point about the TimewarpVAE model working better for lower-dimensional _latent_ spaces (different from the dimensionality of the _input_ and _output_ spaces), we believe this reflects the fact that for larger latent spaces many competing approaches, like PCA, already work well, so there is less room for improvement for our approach.
>
> A key question would be in the type of motion. For trajectories of walking, running, dancing, or acrobatics, for example, where multiple limbs are all moving in a coordinated fashion (but possibly with different timings) we believe that the variations in the spatial styles of movements could be encoded into a low-dimensional spatial latent space. However, arbitrary uncoordinated movements would likely be very hard to model into a low-dimensional space, as uncorrelated movements of different body parts would require a latent space that has size increasing exponentially with the number of limbs.
>
> ## Can we have more direct evidence showing the spatial latent variable and canonical time index can independently control the variations of trajectories along the expected dimensions.
>
> We have now added Figure 10 to show explicitly that the timing latent only affects the timing of the trajectory. Although intuitively, based on the architecture diagram in Figure 2, we know that the temporal latent variables $\Theta$ can only affect the time warping and can therefore only affect the timing of the generated trajectory, we agree that it is helpful to show this explicitly in a figure in the paper.
>
> ## The fork dataset has part as the scaled quaternion while the error appears to be evaluated in the sense of Euclidean. Will this impact training and evaluation?
>
> We have now clarified in the paper the important point that we did preprocess the quaternion representations before feeding them to the model to ensure that they are near each other in $\mathbb R^4$. This could have been a problem since (q,x,y,z) and (-q,-x,-y,-z) represent the same orientation. Because the quaternions are near each other, we did not notice issues with using the Euclidean metric. In general, we believe this is because our reconstructions were close enough to the correct rotations that the Euclidean metric was a good proxy for the difference in orientation between the quaternions.
>
>
> ## Can TimewarpVAE work with multiple data modalities, e.g. handwriting trajectories for all alphabetical letters? Will it need prior other than isotropic Gaussian? Will the identification of time variation help us to have a more structured latent space to group each data modal?
>
> This is a very interesting question. For the purpose of this paper, we were primarily interested in building a generative model that could be used by a robot to create new trajectories. We agree with the reviewer that if we were to build a classification model, it may be beneficial to have a more structured prior other than an isotropic Gaussian in order to beat current state-of-the-art.
>
> ## How TimewarpVAE is related to other generative models with dynamical latent space, such as VAE-DMP [a] ?
> We have now added this work to our related work section. The main difference between our work and that work is in the meaning encoded by the values in the latent space. In VAE-DMP, an individual latent vector represents a single state, and a trajectory is formed by a sequence of different states in the latent space. So, that work is able to compress information about individual states into the latent space. Our work, by contrast, compresses information about the whole trajectory into the latent vector. So one point in our latent space can generate a full canonical trajectory (consisting of 200 different positions in the trajectory).
>
> ## Conclusion
> The authors would again like to thank this reviewer for their questions and especially for the revisions we were inspired to make to our paper because of them, including adding further details on our data processing, and adding an important figure to demonstrate the effect of the time warping latent variables.

---

> > ### Comment · Reviewer_ui6D · 2023-11-22
> > **Thanks for the authors' rebuttal**
> >
> > I would like to thank the authors' response to the reviews. The new results on showing the variation on temporal dimension are great to see the model indeed somehow learned to decouple the temporal and spatial variations. This enhances my belief that the paper is above the acceptance threshold.
> >
> > The paper could be stronger by considering more realistic data with high-dimensional input. The encoder part deals with the entire sequence while the decoder is tasked with generating low-dim coordinates with time-index contexts. It would be more convincing to see the input/frame dimension that the model can handle is not limited to 2/3d.
> >
> > I choose to stick to my rate of marginally above the acceptance threshold.

---

### Official Review · Reviewer_sdX1 · 2023-10-31

**Soundness:** 3 good
**Presentation:** 4 excellent
**Contribution:** 2 fair
**Rating:** 5
**Confidence:** 3

**Summary:**

This manuscript proposes a manifold learning technique to parameterize variation in spatial trajectory datasets by separately factorizing spatial and temporal variation. They demonstrate applications to handwriting and fork movement. They propose a fully differentiable architecture that inputs a trajectory and a timepoint in separate branches. The trajectory is passed through a beta VAE and a piece-wise linear time warping module, along with the temporal input, that uses DTW to align the input time to a canonical time. This design attempts to factorize spatial variations to the beta-VAE and temporal variations to the temporal module.  They evaluate their approach by benchmarking the reconstruction error using three latent dimensions against a beta VAE with no time warping and also the timewarpVAE architecture with the timewarping module set to the identity, finding improved millimeter error performance and rate distortion, which is defined as the KL divergence loss in the VAE. They further show improved reconstruction for handwriting datasets compared to PCA, across choices of the latent dimension, and close with some ablations

**Strengths:**

* The model is well described and I believe novel.

* The problem of aligning trajectories across datasets is important, and the latent space examples are nice.

* There is a comprehensive supplement with methodological documentation.

**Weaknesses:**

* I view this paper as borderline because the number of evaluations is low, and more importantly there are no comparisons with other techniques present in the literature, only reduced versions of the model presented. It is unclear exactly how to position the work to the literature. If the contribution is just a way to build generative models of a set of sequences then the contribution is modest, at least with the range of examples shown. If the method surpasses other approaches for sequence reconstruction then it is more valuable, but it would need to be compared to other reconstruction approaches, eg DMP.

* A bit too much background given on the beta-VAE, which is not novel.

* The fork experiment is a bit idiosyncratic and I am not sure what the significance is.

**Questions:**

* Can you demonstrate that the handwriting representation is more useful for a downstream task, for example classification.

* Can you give any quantitative comparisons with DMP style dimensionality reduction?

*

---

> ### Author Response · Authors · 2023-11-18
>
> ## Introduction
> The authors would like to thank reviewer sdX1 for their helpful review. Based on their feedback we have now added an empirical comparison to Dynamic Movement Primitives and have shown that our method outperforms Parametric DMPs on this task. We note that the purpose of the fork experiment is to contextualize this work by learning a generative model that robots could use to execute useful trajectories, for example picking up food off of a plate to assist someone to eat.
>
> ## A bit too much background given on the beta-VAE, which is not novel
> We have condensed the introductory section on beta-VAE and moved the figure of the beta-VAE architecture along with its optimization objective  to Appendix A.1.
>
> ## DMP style dimensionality reduction
> Dynamic Movement Primitives (DMPs) are able to write trajectories in terms of control parameters. This is very useful for changing trajectories during execution and for motion feedback in changing environments. However, we find that building a latent variable model on those control parameters does not build a positional model of a set of trajectories as accurately as when the model is built directly on the positional data (instead of on DMP control parameters). We have added this quantitative comparison to our paper in Figure 5.
>
> ## Classification
> Classification is an interesting challenge. In this work, the primary focus and consideration was on building an accurate generative model to create trajectories that could be executed on a real robot. The general architecture of VAE provides us this ability to generate new trajectories, and since this is our primary goal, we compare the reconstruction accuracy of the generated trajectories. For classification accuracy, it would likely be beneficial to use a model that is tailored specifically to handle discrete classes of data. As mentioned by reviewer ui6D, training on multiple classes would likely benefit from a generative model with a more structured prior in order to outperform state of the art.
>
> ## Conclusion
> We would like to thank the reviewer for this helpful review, as it has inspired us to add a quantitative comparison to Parametric DMP, showing that our approach outperforms this baseline method.

---

### Meta-Review · Area_Chair_rzLr · 2023-12-19

**Metareview:**

The paper presents TimewarpVAE, a technique for factorizing spatial and temporal variations in trajectory datasets. Reviewers appreciated the paper's clear exposition and detailed experiments. However, there were concerns about the novelty in light of related work, limited comparisons with existing literature, and lack of more complex problems. I do not recommend acceptance at this time. I encourage the authors to demonstrate effectiveness on more complex, high-dimensional problems and further clarify the novelty of the approach compared to similar works in the field.

**Justification For Why Not Higher Score:**

Relative lack of novelty and evaluation on complex problems.

**Justification For Why Not Lower Score:**

N/A

---

### Decision · Program_Chairs · 2024-01-16

Reject